# DeepAGREL: Biologically plausible deep learning via direct reinforcement

## Abstract

While much recent work has focused on biologically plausible variants of error-backpropagation, learning in the brain seems to mostly adhere to a reinforcement learning paradigm; biologically plausible neural reinforcement learning frameworks however were limited to shallow networks learning from compact and abstract sensory representations. Here, we show that it is possible to generalize such approaches to deep networks with an arbitrary number of layers. We demonstrate the learning scheme – DeepAGREL – on classical and hard image-classification benchmarks requiring deep networks, namely MNIST, CIFAR10, and CIFAR100, cast as direct reward tasks, both for deep fully connected, convolutional and locally connected architectures. We show that for these tasks, DeepAGREL achieves an accuracy that is equal to supervised error-backpropagation, and the trial-and-error nature of such learning imposes only a very limited cost in terms of training time. Thus, our results provide new insights into how deep learning may be implemented in the brain.

## 1 Introduction

Among the learning rules for neural networks, reinforcement learning (RL) has the important virtue of occurring in animals and humans. Hence, RL by artificial neural networks can be used as a model for learning in the brain (Bishop et al., 1995). Indeed, previous theories have suggested how powerful RL rules inspired by artificial neural networks could be implemented in the brain (Roelfsema & Holtmaat, 2018) and the methodology for shaping neural networks with rewards and punishments is an active area of research (Schmidhuber et al., 2011; Friedrich et al., 2010; Vasilaki et al., 2009; O'Reilly & Frank, 2006; Huang et al., 2013).

In current deep learning, deep artificial neural networks are typically trained using supervised learning, with variants of the error-backpropagation (EBP) rule. EBP is a method that adjusts synaptic weights in multilayer networks to reduce the errors in the mapping of inputs into the lower layer to outputs in the top layer. It does so by first computing the output error, which is the difference between the actual and desired activity levels of output units, and then determines how the strength of connections between successively lower layers should change to decrease this error using gradient descent (Rumelhart et al., 1986).

Can the brain, with its many layers between input and output indeed solve this credit-assignment problem in a manner that is as powerful as deep learning? Similarly to deep neural networks, the brain of humans and animals are composed of many layers between the sensory neurons that register the stimuli and the motor neurons that control the muscles. Hence it is tempting to speculate that the methods for deep learning that work so well for artificial neural networks also play a role in the brain (Marblestone et al., 2016; Scholte et al., 2017). A number of important challenges need to be solved, however, and some of them were elegantly expressed by Francis Crick who argued that the error-backpropagation rule is neurobiologically unrealistic (Crick, 1989). The main question is: how can the synapses compute the error derivative based on information available locally? In more recent years, researchers have started to address this challenge by proposing ways in which learning rules that are equivalent to error-backpropagation might be implemented in the brain (Urbanczik & Senn, 2014; Schiess et al., 2016; Roelfsema & Ooyen, 2005; Rombouts et al., 2015; Brosch et al., 2015; Richards & Lillicrap, 2019; Scellier & Bengio, 2019; Amit, 2018; Sacramento et al., 2018), most of which were reviewed in (Marblestone et al., 2016).

An important question is whether supervised learning, where a potentially high-dimensional computed outcome is compared to a target outcome of equal dimensionality, is a biologically plausible paradigm for animal learning (Marblestone et al., 2016): while examples of imitation learning exist, most animal learning would be classified as either unsupervised, self-supervised or of a reinforcement learning nature. Plausible deep learning approaches based on reinforcement learning however are lacking.

Here we provide the first RL approach to deep biologically plausible learning that compares favorably performance-wise to supervised learning results found in the (recent) literature (Amit, 2018; Scellier & Bengio, 2019). Using a direct-reward paradigm, we show how this framework can be linked precisely to error backpropagation, with q-values updated after every single action. We then show that this approach can be successfully applied to deep networks for image classification tasks.

As a starting point, we focus on a particular type of biologically plausible learning rule for shallow networks known as AGREL (attention-gated reinforcement learning, reviewed in Richards et al. (2019)) and AuGMEnT (attention-gated memory tagging) (Roelfsema & Ooyen, 2005; Rombouts et al., 2015). These learning rules exploit the fact in a RL setting the synaptic error derivative can be split into two factors: a reward prediction error which is positive if an action selected by the network is associated with more reward than expected, or, if the prospects of receiving reward increase while it is negative, if the outcome of the selected action is disappointing. In the brain, the RPE is signaled by neuromodulatory systems that project diffusely to many synapses so that they can inform them about the RPE (Schultz, 2002); the second factor is an attentional feedback signal that is known to propagate from the motor cortex to earlier processing levels in the brain (Roelfsema & Holtmaat, 2018; Pooresmaeili et al., 2014). When a network chooses an action, this feedback signal is most pronounced for those neurons and synapses that can be held responsible for the selection of this action and hence for the resulting RPE. These two factors jointly determine synaptic plasticity. As both factors are available at the synapses undergoing plasticity, it has been argued that learning schemes such as AGREL and AuGMEnT are indeed implemented in the brain (Roelfsema & Holtmaat, 2018). However, the AGREL and AuGMEnT frameworks have only been applied to networks with a single hidden layer, and modeled tasks with only a handful input neurons.

The present work has two contributions. The first is the development of AGREL/AuGMEnT-derived biologically realistic learning rules for deep networks composed of multiple hidden layers in a RL setting: DeepAGREL. The second is the comparison of the efficacy of trial-and-error learning to supervised learning with EBP, in challenging problems. We investigated how DeepAGREL copes with different datasets, specifically MNIST, CIFAR10 and CIFAR100, trained as direct reward RL tasks. This learning rule is mathematically equivalent to a version of error backpropagation in a reinforcement learning setting that trains one output unit at a time selected by an $\epsilon$-greedy mechanism. We show that it achieves an accuracy that is essentially equivalent to EBP, at a very limited cost in terms of training time (1.5 to 2 times slower), which is caused by the trial-and-error nature of RL.

## 2 BIOLOGICALLY PLAUSIBLE DEEP LEARNING THROUGH REINFORCEMENT

We here generalize and extend AGREL (Roelfsema & Ooyen, 2005; Rombouts et al., 2012b) to networks with multiple layers. To achieve this, we assume that network nodes correspond to cortical columns which feedforward and feedback input (Roelfsema & Holtmaat, 2018; Scellier & Bengio, 2019). In the present implementation we use a feedforward neuron and a feedback neuron per node, shown as blue and green circles in **Fig. 1**. Neurobiological findings suggest that feedforward activation of neurons gates feedback in the brain so that the activity of neurons that are not well driven by feedforward input are also hardly influenced by feedback connections (Van Kerkoerle et al., 2017; Roelfsema, 2006; Treue & Trujillo, 1999): we integrate such a gating mechanism in our learning rule (see below), and use rectified linear (ReLU) functions as activation function of the neurons in the network. This simplifies the learning rule, because the derivative of the ReLU is equal to zero for negative activation values, and has a constant positive value for positive activation values. Note however that the presented learning rule easily generalizes to other activation functions.

The network aims to select as an "output action" the correct class for each image. Each computed output value is treated as a direct reward prediction (Q-value) and one class is selected. For this selected class $s$ (and only for this class $s$) a reward $r$ is assigned and a reward prediction error $\delta$ is

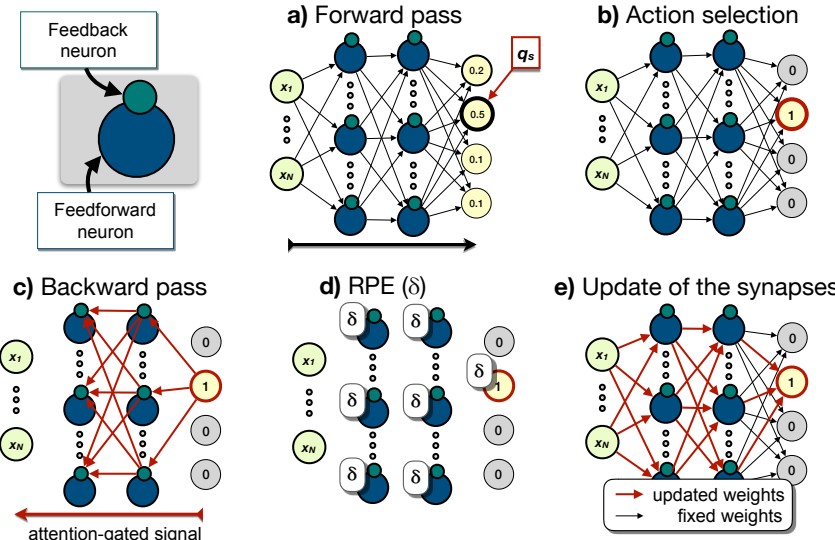

**Figure 1:** Schematic depiction of DeepAGREL. At each node, a feedforward neuron (blue) and a feedback neuron (green) are present; separate feedforward and feedback weights connect the nodes in the network.

computed as the difference between the predicted reward and the actual reward, $\delta - Q_s$. In the brain, this reward prediction error would correspond to the global release of dopamine (Schultz, 2002).

There are five phases upon presentation of an input image: the signal is propagated through the network by feedforward connections to obtain activations for the output units where the Q-values are computed (**Forward pass**, **Fig. 1 a**), in the output layer one output unit wins in a stochastic, competitive action selection process (**Action selection**, **Fig. 1 b**), the selected output unit causes (attention-like) feedback to the feedback unit of each node (**Backward pass**, **Fig. 1 c**, note that this feedback network propagates information about the selected action, just as in the brain, see e.g. (Roelfsema & Holtmaat, 2018), rather than explicitly propagating error signals). A reward prediction error $\delta$ is globally computed (**Fig. 1 d**) after the outcome of the action is evident, and the strengths of the synapses (both feedforward and feedback) are updated (**Fig. 1 e**).

The proposed learning rule, DeepAGREL, has four factors:

$$\Delta w_{i,j} = pre_i \cdot post_j \cdot \delta \cdot fb_j \,, \tag{1}$$

where $\Delta w_{i,j}$ is the change in the strength of the synapse between units $i$ and $j$, $pre_i$ is the activity of the presynaptic unit, $post_j$ the activity of the postsynaptic unit and $fb_j$ the amount of feedback from the selected action arriving at feedback unit $j$ through the feedback network. This local learning rule governs the plasticity of both feedforward and feedback connections between the nodes.

We first consider learning by a network with two fully connected hidden layers comprised of ReLU units (as in **Fig. 1**), and we then derive how the proposed learning scheme can train networks with an arbitrary number of layers in a manner that provides synaptic changes that are equivalent to a particular form of error-backpropagation.

In the network with two hidden layers, there are $N$ input units with activities $x_i$. The activation of the $J$ neurons in the first hidden layer, $y_j^{(1)}$, is given by:

$$y_j^{(1)} = \text{ReLU}\left(a_j^{(1)}\right) \qquad \text{with} \quad a_j^{(1)} = \sum_{i=1}^{N} u_{i,j} \cdot x_i \,, \tag{2}$$

where $u_{i,j}$ is the synaptic weight between the $i$-th input neuron and the $j$-th neuron in the first hidden layer, and the ReLU function can be expressed as: $\text{ReLU}(x) = max(0, x)$.

Similarly, the activations of the $K$ neurons in the second hidden layer, $y_k^{(2)}$, are obtained as follows:

$$y_k^{(2)} = \text{ReLU}\left(a_k^{(2)}\right) \qquad \text{with} \quad a_k^{(2)} = \sum_{j=1}^{J} v_{j,k} \cdot y_j^{(1)}, \tag{3}$$

with $v_{j,k}$ as synaptic weight between the $j$-th neuron in the first hidden layer and the $k$-th neuron in the second hidden layer. The $L$ neurons in the output layer are fully connected (by the synaptic weights $w_{k,l}$) to the second hidden layer and will compute a linearly weighted sum of their inputs:

$$q_l = \sum_{k=1}^{K} w_{k,l} \cdot y_k^{(2)}, \tag{4}$$

which we treat as Q-values as defined in Reinforcement Learning (Sutton et al., 1998), from which actions (or classifications) are selected by an action selection mechanism.

For the action-selection process, we implemented a max-Boltzmann controller (Wiering & Schmidhuber, 1997): the network will select the output unit with the highest Q-value as the winning unit with probability $1 - \epsilon$, and otherwise it will probabilistically select an output unit using a Boltzmann distribution over the output activations:

$$\text{P}(z_l = 1) = \frac{\exp q_l}{\sum_l \exp q_l}. \tag{5}$$

After the competitive action selection process, the activity of the winning unit $s$ is set to one and the activity of the other units to zero, i.e. $z_{l=s} = 1$ and $z_{l \neq s} = 0$. The network then receives a scalar reward $r$ and a globally available RPE $\delta$ is computed as $\delta = r - q_s$, where $q_s$ is the activity of the winning unit (see **Fig. 1**), i.e. the current estimate of reward expectancy, which seems to be coded for in the brain (Schultz, 2002), leading to a global error $E = \frac{1}{2}\delta^2$. In a classification task, we set the direct reward $r$ to 1 when the selected output unit corresponds to the correct class, and we set the reward to 0 otherwise.

Next, only the winning output unit starts propagating the feedback signal – the other output units are silent. This feedback passes through the feedback connections with their own weights $w'$ to the feedback neurons in the next layer, where the feedback signal is gated by the local derivative of the activation function, and then further passed to the next layer of feedback neurons through weights $v'$, and so on. Hence only neurons that receive bottom-up input participate in propagating the feedback signal. We will demonstrate that this feedback scheme locally updates the synapses of the network in a manner equivalent to a particular form of error-backpropagation.

Given the learning rate $\alpha$, the update of the feedforward weights $w_{k,s}$ between the last hidden layer and the selected unit $s$ in the output layer (but the same rule holds for the corresponding feedback weights, indicated as $w'_{s,k}$) is given by:

$$\Delta w_{k,s} = \alpha \cdot \delta \cdot y_k^{(2)} \cdot z_s = \Delta w'_{s,k} \quad \text{and} \quad \Delta w_{k,l \neq s} = 0 = \Delta w'_{l \neq s,k}. \tag{6}$$

The feedforward and feedback weights $v$ and $v'$ between the first and second hidden layer change as follows:

$$\Delta v_{j,k} = \alpha \cdot \delta \cdot y_j^{(1)} \cdot g_{(2)_k} \cdot w'_{s,k} \cdot z_s = \alpha \cdot \delta \cdot y_j^{(1)} \cdot g_{(2)_k} \cdot fb_{y_k^{(2)}} = \Delta v'_{k,j}, \tag{7}$$

$$\text{with} \quad g_{(2)_k} = \begin{cases} 1 & \text{if} \quad y_k^{(2)} > 0, \\ 0 & \text{otherwise}, \end{cases} \tag{8}$$

$$fb_{y_k^{(2)}} = \sum_l g_{(O)_l} \cdot w'_{l,k} \cdot z_l = w'_{s,k} \cdot z_s, \tag{9}$$

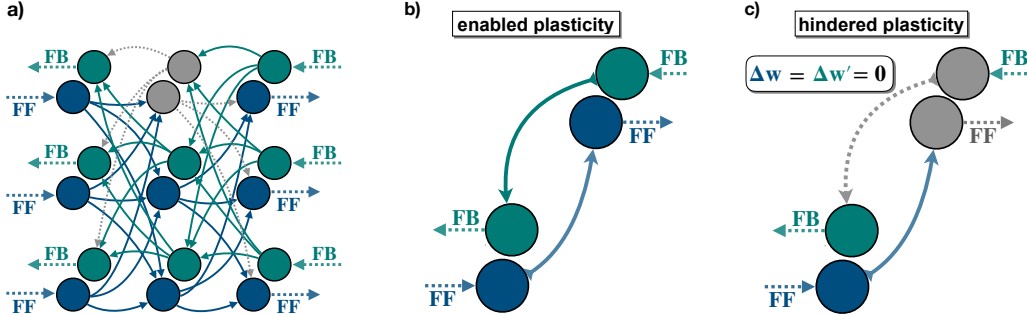

**Figure 2:** DeepAGREL algorithm plasticity gating. **a)** Example hidden layers of a network; **b)** when the activity of the feedforward neuron is above the threshold, the feedback signal is propagated to lower neurons and plasticity is enabled; **c)** when the input to the feedforward unit stays below the threshold for activation the feedback signal is not propagated to the lower layer and plasticity is disabled.

where $fb_{y_k^{(2)}}$ is the feedback coming from the selected output $(O)_l$, and $g_{()}$ denotes the feedback gating. The weights $u$ between the inputs and the first hidden layer are then adapted as:

$$\Delta u_{i,j} = \alpha \cdot \delta \cdot x_i \cdot g_{(1)_j} \cdot \sum_k v'_{k,j} \cdot g_{(2)_k} \cdot w'_{s,k} \cdot z_s$$
$$= \alpha \cdot \delta \cdot x_i \cdot g_{(1)_j} \sum_k v'_{k,j} \cdot g_{(2)_k} \cdot fb_{y_k^{(2)}} = \alpha \cdot \delta \cdot x_i \cdot g_{(1)_j} \cdot fb_{y_j^{(1)}} ,$$

(10)

$$\text{with} \quad g_{(1)_j} = \begin{cases} 1 & \text{if} \quad y_j^{(1)} > 0 \,, \\ 0 & \text{otherwise} \,, \end{cases}$$

(11)

$$fb_{y_j^{(1)}} = \sum_k g_{(2)_k} \cdot v'_{kj} \cdot fb_{y_k^{(2)}} ,$$

(12)

which is the feedback coming from the second hidden layer. $fb_{y_k^{(2)}}$ and $fb_{y_j^{(1)}}$ represent the activity of feedback neurons $y_k^{(2)}$ and $y_j^{(1)}$, which are activated by the propagation of signals through the feedback network, once an action has been selected.

For deeper networks, updates of feedforward synapses $\Delta w_{p,m}$ from $p$-th neuron in the $n$-th hidden layer onto $m$-th feedforward neuron in the $(n+1)$-th hidden layer are thus computed as:

$$\Delta w_{p,m} = \alpha \cdot \delta \cdot y_p^{(n)} \cdot g_{(n+1)_m} \cdot fb_{y_m^{(n+1)}} ,$$

(13)

and are equal to the update of the corresponding feedback synapses $\Delta w'_{m,p}$, where the activity of the feedback unit is determined by the feedback signals coming from the $(n+2)$-th hidden layer:

$$fb_{y_m^{(n+1)}} = \sum_q g_{(n+2)_q} \cdot v'_{q,m} \cdot fb_{y_q^{(n+2)}} ,$$

(14)

with $q$ indexing the units of the $(n+2)$-th hidden layer.

The update of a synapse is thus expressed as the product of four factors: the RPE $\delta$, the activity of the presynaptic unit, the activity of postsynaptic feedforward unit and the activity of feedback unit of the same postsynaptic node, as in Eq. 1, of which eqs. (6), (7) and (10) are all variants. Notably, all the information necessary for the synaptic update is available locally, at the synapse. Moreover, simple inspection shows tat the identical update for both feedforward and corresponding feedback synapses (i.e., $\Delta w_{k,l}$ and $\Delta w'_{l,k}$, $\Delta v_{j,k}$ and $\Delta v'_{k,j}$, and $\Delta u_{i,j}$ and $\Delta u'_{j,i}$) can be computed locally.

The role of the feedback units in each node is to gate the plasticity of feedforward connections (as well as their own plasticity): $fb_j$ acts as a plasticity-gating term, which determines the plasticity of synapses onto the feedforward neuron. In the opposite direction, the feedforward units gate the activity of the feedback units. In **Fig. 2** examples of these interactions are illustrated. Feedback gating

$g_j$ is shaped by the local derivative of the activation function, which, for a unit with a ReLU activation function, corresponds to an all-or-nothing gating signal: for ReLU feedforward units, the associated feedback units of a node are only active if the feedforward units are activated above their threshold (**Fig. 2b**), otherwise the feedback units remain silent and they do not propagate the feedback signal to lower processing levels (**Fig. 2c**).

There is neuroscientific evidence for the gating of plasticity of feedforward connections by the activity of feedback connections (Roelfsema & Holtmaat, 2018). Gating of the activity of feedback units by the activity of feedforward units is also in accordance with neurobiological findings: attentional feedback effects on the firing rate of sensory neurons are pronounced if the neurons are well driven by a stimulus and much weaker if they are not (Van Kerkoerle et al., 2017; Roelfsema, 2006; Treue & Trujillo, 1999). The distinction between feedforward and feedback nodes and the implemented gating mechanisms makes the update rule local at any depth within the network.

DeepAGREL is equivalent to a special form of error-backpropagation (see Appendix A for the complete derivation):The standard error-backpropagation equations for a weight between units $p$ and $m$ in layer $n$ and $n + 1$ respectively are:

$$\Delta w_{p,m} = -\alpha y_p^{(n)} y_m^{(n+1)'} e_m^{(n+1)}, \quad \text{with} \quad e_m^{(n+1)} =: \frac{\partial E}{\partial y_m^{(n+1)}} = \sum_q w_{m,q} y_q^{(n+2)'} e_q^{(n+2)}, \quad (15)$$

with $q$ indexing the units of the $(n + 2)$-th hidden layer and $y_{<\cdot>}^{(\cdot)'}$ indicating the derivative of $y_{<\cdot>}^{(\cdot)}$. This corresponds to the DeepAGREL equations for the adjustment to the winning output when we set $y_j^{()'} = g_j$ and $e_{l=s}^{(O)} = \frac{\partial E}{\partial q_s} = -\delta$, and $e_{l\neq s}^{(O)} = 0$:

$$\Delta w_{p,m} = \alpha \cdot \delta \cdot y_p^{(n)} \cdot g_m \cdot fb_{y_m^{(n+1)}}. \quad (16)$$

Compared to error-backpropagation, in the RL formulation of DeepAGREL only the error $e_l$ for the winning action $l$ is non-zero, and the weights in the network are adjusted to reduce the error for this action only. Depending on the action-selection mechanism, this trial-and-error approach will adjust the network towards selecting the correct action, while the Q-values for incorrect actions will only decrease in strength occasionally, when they are chosen. This contrasts standard error-backpropagation, which will continuously drive the values of the incorrect actions to lower action values. Hence, RL of Q-values by DeepAGREL is expected to be slower than learning with a fully supervised method such as error-backpropagation. We will test these predictions in our simulations.

## 3 EXPERIMENTS

We tested the performance of DeepAGREL on the MNIST, CIFAR10 and CIFAR100 datasets, which are classification tasks, and therefore simpler than more general RL settings that necessitate the learning of a number of intermediate actions before a reward can be obtained. These types of tasks have been addressed elsewhere (Rombouts et al., 2015) with AGREL/AuGMeNT. This previous work used compact and abstracted sensory representations, and the present work therefore addresses much more complex input patterns.

The MNIST dataset consists of 60,000 training samples (i.e. images of 28 by 28 pixels), while the CIFAR datasets comprise 50,000 training samples (images of 32 by 32 pixels, and 3 values for the color at each pixel), of which 1,000 were randomly chosen for validation and left out of the training set. We use a *batch gradient* to speed up the learning process (but the learning scheme also works with learning after each trial, i.e. not in batches): 100 samples were given as an input, the gradients were calculated, divided by the batch size, and then the weights were updated, for each batch until the whole training dataset was processed (i.e. for 590 or 490 batches in total), indicating the end of an *epoch*. At the end of each epoch, a validation accuracy was calculated on the validation dataset. An early stopping criterion was implemented: if for 20 consecutive epochs the validation accuracy had not increased, learning was stopped.

We ran the same experiments with DeepAGREL and with error-backpropagation for neural networks with with three and four hidden layers, as in (Amit, 2018). The first layer could be either convolutional or locally connected (ie, with untied weights), the second layer was convolutional but with a stride of

2 in both dimensions, to which a dropout of 0.8 (i.e. 80% of the neurons in the layer were silent) was applied, then two fully connected layers followed (with the last layer having a dropout rate of 0.3). At the level of the output layer (which had 10 neurons for MNIST and CIFAR10, and 100 for CIFAR100) for error-backpropagation a softmax was applied and a cross-entropy error function was calculated. We decided to test networks with locally connected layers because such an architecture could represent the biologically plausible implementation of convolutional layers in the brain (since shared weights are not particularly plausible, e.g. Cox et al. (2005)). However, since such layers introduce many more parameters in the network, hence we add them here to demonstrate the performance penalty of increasing complexity. Moreover, instead of using max pooling layers to reduce the dimensionality of the layer following the convolutions, we substituted such layers with convolutional layers with equal number of filters and kernel size, but with strides (2,2), as described in (Springenberg et al., 2014). As argued by Hinton (Hinton et al., 2016), dropout is biologically plausible as well: by removing random hidden units in each training run, it simulates the regularisation process carried out in the brain by noisy neurons.

In summary, we ran experiments with the following architectures for the Conv respectively LocCon experiments in Table 1:

a) `conv32 3x3; conv32 3x3 str(2,2); drop.8; full 1,000; full500; drop.3`,

b) `loccon32 3x3; conv32 3x3 str(2,2); drop.8; full 1,000; full500; drop.3`,

with 10 different seeds for synaptic weight initialization. All weights were randomly initialized within the range $[-0.02, 0.02]$ and the feedback synapses were identical to the feedforward synapses (strict reciprocity). For MNIST only we also performed a few experiments with fully connected networks, of which the weights were initialized in $[-0.05, 0.05]$. For CIFAR10 and CIFAR100, we also separately ran the same experiments with 10x standard data augmentation (zooming, shifting, vertical reflections) (Hernández-García & König, 2018).

## 4    RESULTS

**Table 1** presents the results of simulations with the different learning rules. We trained networks with only three hidden layers and networks four hidden layers; these networks had an extra hidden layer with 1000 units. We used 10 seeds for each network architecture and report the results as *mean (standard deviation)*. Our first result is that DeepAGREL reaches a relatively high classification accuracy of 99.16% on the MNIST task, obtaining essentially the same performance as standard error-backpropagation both with the architectures with convolutions and straightforward fully connected networks. The speed of convergence using DeepAGREL was a factor of 1.5 to 2 slower compared to using error-backpropagation for networks with convolutional layers, while it was a factor of 2.5 slower in networks for locally connected layers, but performing slightly better than error-backpropagation.

The results obtained from networks trained on the CIFAR10 dataset show that networks trained with DeepAGREL reached the same accuracy (if not higher) than with error-backpropagation. The number of epochs required for the networks to meet the convergence criterion was also comparable.

**Table 1** also shows the results obtained from networks trained on CIFAR100. Without data augmentation, the final accuracy obtained with DeepAGREL was somewhat lower than with error-backpropagation. However, DeepAGREL learns the CIFAR100 classification task with a convergence rate only 2 to 2.5 times slower than error-backpropagation and the rate for CIFAR10. Adding in standard data augmentation, both algorithms perform much better, and DeepAGREL closes the gap with error backpropagation in terms of accuracy (also for CIFAR10). These results shows that such trial-and-error learning rule can scale up to a 10 times higher number of classes with a relatively small penalty.

To illustrate the learning process of networks trained with the DeepAGREL reinforcement learning approach, we show how the reward probability increases (**Fig. 3**) during the training, compared to how the error (plotted as 1 - error) evolves throughout the epochs for 10 networks trained with error-backpropagation (in both cases, $mean \pm 2\sigma$ from 10 example networks is plotted as a function of the epochs), both for CIFAR10 (left panel) and CIFAR100 (right panel). Indeed, learning with supervision is faster than learning by trial-and-error, a difference that is particularly pronounced for CIFAR100 where the probability of choosing the right action starts at only 1%.

**Table 1:** Results (averaged over 10 different seeds, the mean and standard deviation are indicated; in some cases - indicated with "*" - only 9 out of 10 seeds converged). For CIFAR10 and CIFAR100, we also applied data augmentation where indicated.

| | Rule | $1^{st}$ layer | Hidden units | $\alpha$ | Epochs [#] | Accuracy [%] |
|---|---|---|---|---|---|---|
| **MNIST** | DeepAGREL | Full | 1500-1000-500 | 5e-01 | 130 (54) | 98.33 (0.09) |
| | Error-BP | Full | 1500-1000-500 | 1e-01 | 92 (11) | 98.32 (0.04) |
| | DeepAGREL | Conv | 21632-5408-1000-500 | 1e+00 | 59 (21) | 99.16 (0.16) |
| | Error-BP | Conv | 21632-5408-1000-500 | 1e-02 | 33 (12) | 99.21 (0.17) |
| | DeepAGREL | LocCon | 21632-5408-1000-500 | 1e+00 | 66 (15) | 98.30 (0.22) |
| | Error-BP | LocCon | 21632-5408-1000-500 | 1e-02 | 24 (13) | 98.73 (0.41) |
| **CIFAR10** | DeepAGREL | Conv | 28800-7200-1000-500 | 1e+00 | 115 (23) | 73.54 (1.35) |
| | Error-BP | Conv | 28800-7200-1000-500 | 1e-03 | 83 (21) | 71.25 (1.08) |
| | DeepAGREL | LocCon | 28800-7200-1000-500 | 1e+00 | 173 (36) | 64.37 (2.41) |
| | Error-BP | LocCon | 28800-7200-1000-500 | 1e-03 | 145 (16) | 64.65 (1.16) |
| **DATA AUGM** | DeepAGREL | Conv | 28800-7200-1000-500 | 1e+00 | 64(18) | 79.97(1.09) |
| | Error-BP | Conv | 28800-7200-1000-500 | 1e-03 | 47(23) | 78.39(0.67) |
| | DeepAGREL | LocCon | 28800-7200-1000-500 | 1e+00 | 78(25) | 70.87(0.70) |
| | Error-BP | LocCon | 28800-7200-1000-500 | 1e-03 | 56(8) | 72.07(0.46) |
| **CIFAR100** | DeepAGREL | Conv | 28800-7200-1000-500 | 1e+00 | 230 (30) | 34.90 (1.49)* |
| | Error-BP | Conv | 28800-7200-1000-500 | 1e-03 | 104 (24) | 36.79 (1.78) |
| | DeepAGREL | LocCon | 28800-7200-1000-500 | 1e+00 | 343 (68) | 29.39 (2.38) |
| | Error-BP | LocCon | 28800-7200-1000-500 | 1e-03 | 156 (13) | 32.73 (0.78) |
| **DATA AUGM** | DeepAGREL | Conv | 28800-7200-1000-500 | 1e+00 | 93(18) | 42.98(1.78) |
| | Error-BP | Conv | 28800-7200-1000-500 | 1e-03 | 52(16) | 41.30(0.61) |
| | DeepAGREL | LocCon | 28800-7200-1000-500 | 1e+00 | 123(25) | 37.32(0.76) |
| | Error-BP | LocCon | 28800-7200-1000-500 | 1e-03 | 46(12) | 38.66(0.49) |

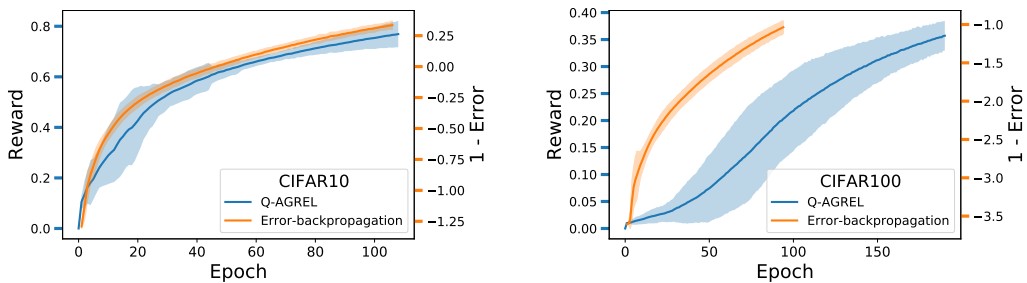

**Figure 3:** Example of learning process with the two learning schemes for CIFAR10 and CIFAR100, without data augmentation.

## 5 DISCUSSION

We implemented a deep, biologically plausible RL scheme called DeepAGREL and found that it was able to train networks to perform the MNIST, CIFAR10 and CIFAR100 tasks as direct reward problems with performance that was nearly identical to error-backpropagation, in particular when we included simple data augmentation. We also found that the trial-and-error nature of learning to classify with RL incurred a very limited cost of 1-2.5x more training epochs to achieve the stopping criterion, even for classifying objects in 100 classes.

The results were obtained with relatively simple network architectures (i.e. not very deep) and learning rules (no optimizers were used except for data augmentation where indicated). Such additions would almost certainly further increase the final accuracy of the DeepAGREL learning scheme.

The present results demonstrate how deep learning can be implemented in a biologically plausible fashion in deeper networks and for tasks of higher complexity by using the combination of a global RPE and "attentional" feedback from the response selection stage to influence synaptic plasticity. Importantly, both factors are available locally, at many, if not all, relevant synapses in the brain (Roelfsema & Holtmaat, 2018). We used symmetrical weights and did not address a potential weight-transport problem (Bartunov et al., 2018) since, as argued in Akrout et al. (2019), given reciprocal learning and weight decay specificity for feedforward and feedback weights emerges.

We demonstrated that DeepAGREL is equivalent to a version of error-backpropagation that only updates the value of the selected action. DeepAGREL was developed for feedforward networks and for classification tasks where feedback about the response is given immediately after the action is selected. However, the learning scheme is a straightforward generalization of the AuGMeNT framework (Rombouts et al., 2012a; 2015), which also deals with RL problems for which a number of actions have to be taken before a reward is obtained. In future work, we aim to to develop more architectures compatible with DeepAGREL and train more complex networks as required for larger tasks like ImageNet-classification. Similarly, as stressed by Richards et al. (2019), theory is needed to quantify the variance of the AGREL approach.

We find it encouraging that insights into the rules that govern plasticity in the brain are compatible with some of the more powerful methods for deep learning in artificial neural networks. These results hold promise for a genuine understanding of learning in the brain, with its many processing stages between sensory neurons and the motor neurons that ultimately control behavior.

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

## A   APPENDIX DERIVATION OF ERROR BACKPROPAGATION

For error-backpropagation in the same networks with error $E$ computed as the summed square error over all output Q-values $q_l$ and target outputs $\hat{q}_l$, $E = -\frac{1}{2}\sum_l (q_l - \hat{q}_l)^2$, if we define $\frac{\partial E}{\partial q_l} = (q_l - \hat{q}_l) := e_l^{(O)}$, where the superscript $(O)$ stands for output layer, the–standard–equations for the synaptic updates are:

$$\Delta w_{k,l} = -\alpha y_k^{(2)} e_l^{(O)} \,, \tag{17}$$

$$\Delta v_{j,k} = -\alpha y_j^{(1)} y_k^{(2)'} \underbrace{\sum_l w_{l,k} e_l^{(O)}}_{} = -\alpha y_j^{(1)} y_k^{(2)'} \underline{e_k^{(2)}} \,, \tag{18}$$

$$\Delta u_{i,j} = -\alpha x_i y_j^{(1)'} \underbrace{\sum_k v_{k,j} y_k^{(2)'} \sum_l w_{l,k} e_l^{(O)}}_{} = -\alpha x_i y_j^{(1)'} \underbrace{\sum_k v_{k,j} y_k^{(2)'} e_k^{(2)}}_{} = -\alpha x_i y_j^{(1)'} \underline{e_j^{(1)}} \,, \tag{19}$$

and in general, for a weight between units $p$ and $m$ in layer $n$ and $n+1$ respectively are:

$$\Delta w_{p,m} = -\alpha y_p^{(n)} y_m^{(n+1)'} e_m^{(n+1)} \,, \qquad \text{with} \quad e_m^{(n+1)} =: \frac{\partial E}{\partial y_m^{(n+1)}} = \sum_q w_{m,q} y_q^{(n+2)'} e_q^{(n+2)} \,, \tag{20}$$

with $q$ indexing the units of the $(n+2)$-th hidden layer and $y_{<>}^{(\cdot)'}$ indicating the derivative of $y_{<>}^{(\cdot)}$. This corresponds to the DeepAGREL equations for the adjustment to the winning output when we set $y_j^{()'} = g_j$ and $e_{l=s}^{(O)} = \frac{\partial E}{\partial q_s} = -\delta$, and $e_{l\neq s}^{(O)} = 0$:

$$\Delta w_{k,l} = \alpha \delta y_k^{(2)} \,, \tag{21}$$

$$\Delta v_{j,k} = \alpha \delta y_j^{(1)} g_k fb_{y_k^{(2)}} \,, \tag{22}$$

$$\Delta u_{i,j} = \alpha \delta x_i g_j \sum_k g_k w_{l,k} fb_{y_k^{(2)}} = \alpha \delta x_i g_j fb_{y_j^{(1)}} \,, \tag{23}$$

and, by recursion,

$$\Delta w_{p,m} = \alpha \delta y_p^{(n)} g_m fb_{y_m^{(n+1)}} \,. \tag{24}$$

