# OpenReview forum: "DeepAGREL: Biologically plausible deep learning via direct reinforcement"
_ICLR.cc/2020/Conference — Reject_

### Official Review · AnonReviewer2 · 2019-10-23
**Official Blind Review #2**

**Rating:** 6

**Review:**

This paper presents DeepAGREL, a framework for biologically plausible deep learning that is modified to use reinforcement learning as a training mechanism. This framework is shown to perform similarly to error-backpropagation on a set of architectures. The idea relies on feedback mechanism that can resemble local connections between real neurons.

This paper is an interesting approach to provide a reinforcement learning paradigm for training deep networks, it is well written and the experiments are convincing, although more explanation about why these specific architectures were tested would be more convincing. I also think the assumptions about feedback connections in real neurons should be visited and more neuroscientific evidence from the literature should be included in the paper. Do we expect feedback to happen at each level of a neuron-neuron interaction and between each pair of connected neurons? Is there a possibility that feedback is more general to sets of neurons, or skips entire layers of neurons? I think more neuroscience background would help this paper (and others on the topic). Nonetheless, I think the paper does offer an interesting proposal of a more biologically plausible form of deep learning.


**Experience Assessment:**

I have read many papers in this area.

**Review Assessment: Checking Correctness Of Derivations And Theory:**

I assessed the sensibility of the derivations and theory.

**Review Assessment: Checking Correctness Of Experiments:**

I assessed the sensibility of the experiments.

**Review Assessment: Thoroughness In Paper Reading:**

I read the paper at least twice and used my best judgement in assessing the paper.

---

> ### Author Response · Authors · 2019-11-10
> **Reply to Reviewer #2**
>
> We selected the same architectures as in [Amit2018] as they were tested on another rule with biologically plausible components, we added this as a reference in the text.
> Regarding feedback connections, for reasons of space, we point to the elaborate review paper by Roelfsema and Holtmaat (2018); we also added a reference to the very recent Richards et al. (2019) paper. The point about the specificity of forward and backward connections however deserves more study. We would argue that given reciprocal learning and weight decay, as argued by Akrout et al. (2019), specificity will emerge. We added this point (and citation) to the discussion.
>
> Richards, Blake A., et al. "A deep learning framework for neuroscience." Nature neuroscience 22.11 (2019): 1761-1770.
> Akrout, M. M., Wilson, C., Humphreys, P., Lillicrap, T., & Tweed, D. (2019). Deep learning without weight transport.

---

### Official Review · AnonReviewer1 · 2019-10-30
**Official Blind Review #1**

**Rating:** 6

**Review:**

Summary:

This paper generalizes the AGREL biologically plausible learning algorithm to deeper networks. It presents experiments to show that the method can scale to medium-sized tasks with only a modest slow down in training speed and essentially no change in classification performance.

Major comments:

This is an interesting paper which shows that a particular biologically plausible learning method can attain comparable performance to gradient descent on small and mid-size visual classification tasks. The proposed mapping to the biology is of a different flavor from many other recently proposed approaches.

It is striking that the CIFAR10 network trains about as fast as the EBP network, while the CIFAR100 network trains much more slowly. This is presumably because randomly guessing the right answer out of 100 possibilities is the bottleneck. The paper could be strengthened by studying the speed of learning as a function of the number of output classes. For this approach to scale to ImageNet, with a 1000-way classification (or to our human visual recognition abilities with far more classes), this is an important scaling dimension to consider.

It should be noted that other biologically plausible schemes like feedback alignment were able to solve CIFAR and other smaller image classification tasks, but struggled when applied to the larger scale ImageNet problem. The paper could be improved by pointing to this limitation of the present work, the possibility that performance could change on larger tasks, and the need to conduct larger experiments in future work.

Personally I think the statement at the beginning of the introduction that only RL occurs in animals and humans is overstated. Unsupervised learning occurs in some form in critical period plasticity, and in various unrewarded statistical learning paradigms, at a minimum.

I would also caution against categorical statements of biological plausibility, for instance, in saying that shared weights in convolutions are biologically implausible (bottom of pg 6). The same criticism has been leveled at gradient descent learning, and as the present submission shows, these intuitive judgements can be misleading.

The distinction between RL and supervised learning is a bit blurred in the present submission because it is considering a classification setting in which exactly one out of a number of possible outputs is rewarded on each trial. This looks very similar to the supervised learning scenario, and relies on stochastic softmax-like competition to select a single output. This approach is very reasonable for mutually exclusive classification tasks, but this is a small subset of the tasks that full EBP could be applied to.

The paper is fairly clear but the explanation of the algorithm could be further streamlined and condensed. Is DeepAGREL equivalent to stochastically selecting a single unit in a softmax layer and then back propagating only its error? It seems so from what I understand, and this may be a straightforward way to explain the algorithm.

Typos:

The text on page 7 describes MNIST performance as 99.17% but the table has 99.16%.



**Experience Assessment:**

I have published one or two papers in this area.

**Review Assessment: Checking Correctness Of Derivations And Theory:**

I assessed the sensibility of the derivations and theory.

**Review Assessment: Checking Correctness Of Experiments:**

I assessed the sensibility of the experiments.

**Review Assessment: Thoroughness In Paper Reading:**

I read the paper at least twice and used my best judgement in assessing the paper.

---

> ### Author Response · Authors · 2019-11-10
> **Reply to Reviewer #1**
>
> Regarding the dependance of learning on the number of classes: we see the cost of “trial-and-error” learning go up from about 150% for MNIST and CIFAR10 to about 200% for CIFAR100, a very modest increase. In our studies, the main problem we encountered when training CIFAR100 instead of CIFAR10 was not so much due to the number of classes, but rather the 10 fold reduced number of samples per class in CIFAR100. For this reason, we added data augmentation.
> Moreover, for what concerns the extension to larger tasks, we demonstrated that the rule works on the standard image classification tasks used for biologically plausible learning, with essentially equivalent performance to error backpropagation. In order to train networks on larger tasks we would need architectures/elements (e.g. max pooling, model ensembles, memory units, etc) for which biologically plausible counterparts still need to be developed in the AGREL setting. This is a goal for future work; we include this point now explicitly in the discussion.
> For the statement in the introduction, there may be some confusion, as we agree: we are followed Yann LeCunn in rephrasing “unsupervised” as “self-supervised learning”. We updated the paper to explicitly include “unsupervised”. Instead, regarding convolutional layers, we are mainly concerned with neuroscientific data that explicitly shows that there are clear instances where vision is not learned in a “position invariant” manner [COX2005]. Clearly, weights are not physically shared in the brain, though one could argue that eye-movements mostly take care of this problem. We added the citation in the paper to be more explicit.
> When stating that this approach is very reasonable for mutually exclusive classification task but that these make only for a small subset of the task that EBP could be applied to the reviewer is correct, thus our statement regarding the nature of learning in animals. At the same time, related work like the cited Rombouts et al (2015) paper, demonstrate that this approach applies to learning very many cognitive tasks. Also see the very recent paper by Richards et al. in Nature Neuroscience (2019), which positions AGREL relative to other approaches (we added this citation in the paper).
> Finally, "Is DeepAGREL equivalent to stochastically selecting a single unit in a softmax layer and then back propagating only its error?" Effectively yes, except that the error is separated from the activity feedback; feeding back the error would be identical. Specifically, DeepAGREL is equivalent to a version of error backpropagation in a reinforcement learning setting that trains one output unit at a time, using an ε greedy mechanism for action selection (i.e. 98% of the times the neuron with the highest activity will be selected). We added a sentence to this effect in the paper (end of the Introduction).
>
> Cox, D. D., Meier, P., Oertelt, N. & DiCarlo, J. J. ‘Breaking’ position-invariant object recognition. Nat. Neurosci. 8, 1145–1147 (2005).
> Richards, Blake A., et al. "A deep learning framework for neuroscience." Nature neuroscience 22.11 (2019): 1761-1770.

---

### Official Review · AnonReviewer4 · 2019-10-30
**Official Blind Review #4**

**Rating:** 1

**Review:**

The paper proposes an backpropagation-free algorithm for training in deep neural networks.
The algorithm called DeepAGREL does not use the chain rule for computing derivatives and instead is based on direct reinforcement.
Authors conduct experiments on MNIST and CIFAR where they find their method to reach performance similar to BP.

Novelty: the method is built upon existing ideas, however, the exact algorithm seems to be novel.

Clarity: although it is possible to understand the algorithm based on the provided textual description, it would not hurt to provide a more formal presentation of the main update equation (1).

It is not clear why authors provide two different expressions for updates in each layer (in terms of fb^{l} and fb^{l+1}).

I would also appreciate a clear discussion on the theoretical properties of the algorithm, for example, is it guaranteed to converge to a critical point of some loss function? Can we derive the update rule from some known optimisation procedure, e.g. REINFORCE?

Quality: Unfortunately I have a number of major issues with this respect.

1. Authors do not clearly pose what are the properties of back propagation that they find biologically non-plausible and how exactly their algorithm is making progress on them. There seems to be some sort of consensus in the literature about these properties and if authors accept it, then it looks like DeepAGREL is not very plausible too, because it does not address the weight-transport issue at all.
2. Many recent results are not even mentioned in the paper, for example, (Bartunov et al, 2019). From not so recent — the whole branch of research around target propagation algorithm (Lee et al, 2014).
3. The experiments are not convincing to me, as the considered network architectures are quite shallow and thus the ability to perform credit assignment can not be demonstrated. Besides that, the locally-connected architecture has only the first layer locally-connected and it does not make much sense to me, as it does not factor out the impact of weight sharing.

Overall, I think the paper is not ready for publication at ICLR.


**Experience Assessment:**

I have published one or two papers in this area.

**Review Assessment: Checking Correctness Of Derivations And Theory:**

I assessed the sensibility of the derivations and theory.

**Review Assessment: Checking Correctness Of Experiments:**

I assessed the sensibility of the experiments.

**Review Assessment: Thoroughness In Paper Reading:**

I read the paper at least twice and used my best judgement in assessing the paper.

---

> ### Author Response · Authors · 2019-11-10
> **Reply to Reviewer #4**
>
> When asking for a more formal presentation of the main update equation, we are not quite clear what the reviewer means: we introduce (1) from a line of research papers, after giving an intuitive explanation, afterwards we show that (1) results in approximate EBP in a trial-and-error fashion.
> Concerning the "two different expressions for updates in each layer" we are not quite sure what the reviewer is referring to, which equations are at issue?
> More theoretical properties of the algorithm: the global error, as indicated in the text (mid page 4), is a MSE: $E = \frac{1}{2}\delta^2 = \frac{1}{2}{(r-q_s)}^2 $. Since it is a stochastic optimisation of this error, we do not have any theoretical guarantees (yet), but, as also pointed out in Richards et al (2019), determining the variance of the approach is important. We added this note in the discussion.
> Indeed we had not addressed the weight-transport issue. As indicated by Tim Lillicrap himself in [Akrout 2019], the key issue is not so much symmetric weights, but being able to locally compute the same error both in the forward and backward network. Having different feedforward and feedback weights but symmetrical updates with weight decay leads to a convergence to the same value, i.e. symmetrical weights. For this reason, we have stuck with symmetrical weights. We added this explicit reference in the text. Of course, we are also arguing that supervised learning is not the proper framework for animal learning.
> Presuming the reviewer is referring to [Bartunov et al 2018], "Assessing the Scalability of Biologically-Motivated Deep Learning Algorithms and Architectures" (NeuriPS2018), when saying that we missed to mention recent results, we remark that this and the idea of Target Propagation is focused on replicating the error-backpropagation learning algorithm in a supervised setting. Bartunov in particular examined Target Propagation and Feedback alignment, showing that both perform substantially worse than error-backprop. Also, while Bartunov et al scale up to ImageNet, the results are so extremely poor that while informative of its failure, we are focusing first on getting much better results compared to EBP in a reinforcement learning setting; to scale to ImageNet size networks, a future goal, we will need to find biologically plausible equivalents for learning through a number of standard deep learning structures, like max-pooling. We note so in the update text.
> And finally, regarding the credit-assignment, we strongly disagree with the reviewer: the results clearly demonstrate proper credit assignment in deep networks, up to 5 layers, as the results correspond to EBP, as the math also demonstrates (and any bug in the code during development also emphatically showed this). For locally connected, we were mostly interested in determining the performance penalty of increasing the complexity; we agree that the architecture used does not completely factor out the impact of weight sharing; and we note so in the text now, we argue however that adding such complexity in terms of weights still demonstrates the effectiveness of the credit-assignment by DeepAGREL.
>
> Akrout, M. M., Wilson, C., Humphreys, P., Lillicrap, T., & Tweed, D. (2019). Deep learning without weight transport.

---

> > ### Comment · AnonReviewer4 · 2019-11-15
> > **Reply**
> >
> > Thank you for your reply.
> >
> > The equation I was referring to is equation (7). After re-reading the manuscript I now see that is the symmetrical learning rules.
> >
> > I appreciate authors reply, however I largely keep to my initial evaluation, mostly because of the clarity issues which raised in my initial review.
> > I am also confused about why the authors did not cite Akrout et al, 2019 and Bartunov et al, 2018 and, more importantly, did not discuss these results in the first place since they seemed to be aware about these papers.
> >
> > My recommendations for future submissions of the manuscript are the following:
> > 1) Fully integrate all past and recent work on biologically-plausible learning and clearly position the proposed framework within the literature.
> > 2) Provide a clear derivation of the learning algorithm, starting from explaining the basic principles (e.g. stochastic optimization, Q-learning, etc) and then use it to derive the update rules.
> > 3) Perform experiments on deeper architectures to confirm the scalability of the proposed method and its ability to perform credit assignment.
> > 4) Improve mathematical notation, reduce the number of potentially confusing elements such as fb_{y_j^{(1)}}.
> >
> > I believe that if the submitted paper indeed possess a great importance for the field, following these recommendations will only make it stronger and increase the chance of acceptance. In the current form I do not find it progressing the field and thus cannot recommend acceptance.

---

### Public Comment · ~Raden_Muaz1 · 2019-10-04
**Wall clock training time?**

Since only neurons connected to selected action neuron are updated for every iteration, does this mean you can save computation time by not computing errors and updates for unrelated neurons? Even though more epochs needed to achieve accuracy similar to error backprop?

How is the whole training algorithm implemented in code? During matmul, what happend to neurons that don't need updates?

How is wall clock training time?

---

> ### Author Response · Authors · 2019-10-07
> **Re: Wall clock training time?**
>
> Thank you for your interest and comment. Yes, theoretically there is a savings in computation time since we do not update non-selected output neurons. This savings however would be extremely small, since all other layers, containing most neurons and connections, are fully updated. So in practise, wall-time per epoch for DeepAGREL and EBP are the same.

---

### Decision · Program_Chairs · 2019-12-19

**Decision:**

Reject

**Comment:**

The paper proposes an RL-based algorithm for training neural networks that is able to match the performance of backprop on CIFAR and MNIST datasets.

The reviewers generally found the algorithm and motivations interesting, but some had issue with the imprecision of the notion of "biologically plausible" used by the authors. One reviewer had issues with missing discussion of related work and also doubts about the meaningfulness of the experiments, since the networks were quite shallow.

For this type of paper, clarity and precision of exposition is crucial in my opinion, and so I recommend rejection at this time, but encourage the authors to take the feedback into account and resubmit to a future venue.